# Sensitivity of Field-Effect Transistor-Based Terahertz Detectors

**DOI:** 10.3390/s21092909

**Published:** 2021-04-21

**Authors:** Elham Javadi, Dmytro B. But, Kęstutis Ikamas, Justinas Zdanevičius, Wojciech Knap, Alvydas Lisauskas

**Affiliations:** 1CENTERA Laboratories, Institute of High Pressure Physics PAS, 01-142 Warsaw, Poland; dbut@mail.unipress.waw.pl (D.B.B.); knap.wojciech@gmail.com (W.K.); 2CEZAMAT, Warsaw Technical University, 02-822 Warsaw, Poland; 3Institute of Applied Electrodynamics and Telecommunications, Vilnius University, LT-10257 Vilnius, Lithuania; kestutis.ikamas@ff.vu.lt (K.I.); justinas.zet@gmail.com (J.Z.); 4Research Group on Logistics and Defense Technology Management, General Jonas Žemaitis Military Academy of Lithuania, LT-10322 Vilnius, Lithuania; 5Laboratoire Charles Coulomb, University of Montpellier, and CNRS, 34095 Montpellier, France

**Keywords:** THz detectors, planar antennas, focal plane arrays, effective antenna area, CMOS detector

## Abstract

This paper presents an overview of the different methods used for sensitivity (i.e., responsivity and noise equivalent power) determination of state-of-the-art field-effect transistor-based THz detectors/sensors. We point out that the reported result may depend very much on the method used to determine the effective area of the sensor, often leading to discrepancies of up to orders of magnitude. The challenges that arise when selecting a proper method for characterisation are demonstrated using the example of a 2×7 detector array. This array utilises field-effect transistors and monolithically integrated patch antennas at 620 GHz. The directivities of the individual antennas were simulated and determined from the measured angle dependence of the rectified voltage, as a function of tilting in the E- and H-planes. Furthermore, this study shows that the experimentally determined directivity and simulations imply that the part of radiation might still propagate in the substrate, resulting in modification of the sensor effective area. Our work summarises the methods for determining sensitivity which are paving the way towards the unified scientific metrology of FET-based THz sensors, which is important for both researchers competing for records, potential users, and system designers.

## 1. Introduction

For a considerably long time, devices that utilise thermal detection principles have been routinely employed as practical detectors in the terahertz (THz) frequency range. The prominent representatives of this class are Golay cells [1], pyroelectric detectors [2], and bolometers [3]. As the radiation can be delivered to the sensing element employing the absorption mechanism, the performance of thermal detectors can be well-calibrated and traced [2]. However, a number of particular characteristics related to thermal detection mechanisms, such as low modulation speed, comparatively large area, and the need for external filters if detector frequency selectivity is required, pose severe limitations for the applicability of such devices. Another class of uncooled THz detectors includes the zero-bias Schottky barrier detector [4,5,6] and the recently emerged family of detectors based on field-effect transistors (FETs). The fast response time, in conjunction with the high sensitivity values of the latter class of devices, makes rectifiers very attractive for practical applications. Furthermore, FET-based detectors profit from integrated circuit technological development and can be produced in large arrays [7,8,9,10,11].

The main principle of efficient THz FET detector operations at frequencies highly exceeding the transistor cut-off limit relies on the excitation of damped plasma oscillations in a two-dimensional gas in the transistor channel [12,13,14,15]. This mechanism was later extended to a device model based on hydrodynamic transport description, in order to account for rectification efficiency and device impedance [16,17,18,19].

It is important to note that, with the introduction of novel detection mechanisms, new challenges for researchers, in terms of determining traceable ways to present their performance values, have appeared. The main challenge arises from the fact that the rectifying part of the FET—an ordinary object of study—is much smaller (often with sub µm dimensions) than the THz wavelength (hundreds of µm). Therefore, the THz radiation has to be coupled to it using a grating or either intentional or unintentional antennae, which defines the effective detector area in the process of sensitivity determination. Compared with two-terminal devices, such as Schottky diodes, which benefit from the developed solutions for integration into metal waveguides, there exist no ready solutions for three-terminal transistors. Therefore, initial proof-of-principle experiments have been performed using discrete FETs [12,13,20]. A report of the efficient detection of 700 GHz radiation using wire-bonded 120 nm CMOS devices [21] has raised exceptional attention. As discrete devices possess no dedicated antennas, the assumption is made that the fraction of the power delivered to the device is proportional to the diffraction-limited spot. However, a comprehensive study performed much later using discrete GaAs devices brought attention to the fact that both the structure of contacts and bonding wires play unintentional roles as antennas. The performance estimates based on the antenna-related effective area and that of the diffraction-limited spot may differ by more than 18 times [22].

The first array, which combined complementary metal–oxide–semiconductor (CMOS) FETs with on-chip integrated antennas, was developed using a 250 nm technology node [14]. The array consisted of 3 × 5 pixels with resonant patch antennas and on-chip integrated amplifiers and reported a responsivity of 70 kV/W and a noise equivalent power (NEP) of 300 pW/$\sqrt{\mathrm{Hz}}$ at 30 kHz chopping frequency. Although these devices contained antennas for which the effective areas could have been simulated using a standard approach, it was assumed that, due to the proximity of neighbouring detectors and concomitant power-sharing, the effective area is set by the pitch between devices. Further optimisations have been performed for the antenna, as well as the device; for example, the employment of 65 nm CMOS SOI technology resulted in an improvement in NEP values to ∼50 pW/$\sqrt{\mathrm{Hz}}$ at 650 GHz [23] or 20 pW/$\sqrt{\mathrm{Hz}}$ at 590 GHz [24] for a 90 nm CMOS process with the assumption of effective areas supported by electromagnetic simulations.

Evaluation of the performance for substrate-lens coupled devices poses another challenge, as the full-wave simulation of structures at THz frequencies requires invoking computer resources that are hardly accessible. One of the first reports on detectors that utilise broadband antennas with preferred coupling direction from the substrate side was presented in [7] for a 32 × 32 pixel array fabricated using 65 nm CMOS technology. Although the reported minimum NEP of ∼100 pW/$\sqrt{\mathrm{Hz}}$ at 856 GHz was slightly larger than that for resonant devices, a bandwidth of 200 GHz was achieved.

Much progress has been reported in terms of improving the optical performance of devices implemented using CMOS technologies. For example, employing biquad antennas allows for reaching ∼25 pW/$\sqrt{\mathrm{Hz}}$ at 1 THz with more than 400 GHz bandwidth [25], or larger than 1 THz bandwidth with ∼50 pW/$\sqrt{\mathrm{Hz}}$ for the bow-tie antenna-coupled detectors [26], which are competitive to commercial quasioptical Schottky diode detectors [27]. In parallel to the progress in silicon technology-based devices, sensitive detectors have been produced using field-effect transistors fabricated using III–V technologies, mono- and bilayer graphene, as well as by invoking other detection principles, such as ballistic transport [28] or thermoelectric currents [29,30,31]. It is also important to note that the presented FET NEP values gradually approach the level of Schottky diodes, with NEP values of a few pW/$\sqrt{\mathrm{Hz}}$ at room temperature. For a more detailed discussion of various types of terahertz detectors, we recommend the following review works [32,33].

One can note that, depending on the purpose of the device, such as designing a single detector; serving as part of an array or as an imaging element; providing optimal point-to-point power delivery; or proof-of-the-principle for a novel structure or detection phenomena, it is often required to apply different methods to specify the main figures-of-merit. This manuscript summarises six main methods reported in the recent literature and addresses the differences between them. Furthermore, it summarises the performance values and characterisation methods used for state-of-the-art devices in three material system-related groups: Silicon, III–V semiconductors, and novel low-dimensional structures. Finally, we present a comprehensive simulation-supported experimental study of devices implemented in a 2 × 7 array using 65 nm CMOS and illustrate the spread of performance values under different methods.

## 2. Commonly Used Methods for the Estimation of THz Detector Area

Based on our review, performed on over 50 state-of-the-art field-effect transistor-based detectors, we outline six methods commonly used to estimate the THz detector area in the following. The summary of methods, with reference to the relative reports, is presented in Table 1.

*From the antenna gain*. Probably the most accepted method in the field of microwave antennas is relating the effective area of the antenna, $A_{\mathrm{eff},I}=G\lambda_{0}^{2}/4\pi$, with its gain *G* and free-space wavelength $\lambda_{0}$ [38] [p. 1–10]. This method has been described in textbooks and simulation tools. The antenna gain and directivity *D* are directly related by the antenna efficiency η (i.e., G=ηD). If the intensity, $I_{\mathrm{THz}}$, of radiation at the plane of the detector is known—for example, when performing direct measurements or estimating it using the gain of the antenna used at the THz source side—one can estimate the power which is delivered into the detector circuit as $P_{D}=A_{\mathrm{eff},I}I_{\mathrm{THz}}$. As a matter of fact, it is essential to estimate $P_{D}$, as well as the power matching coefficient and other related quantities, in order to perform a quantitative comparison between the analytical CAD-based simulation, which usually starts from the intrinsic device or phenomenon description, and the experiment; however, in contrast to microwave frequency range-related instrumentation, at present, one can hardly detach the THz rectifying transistor from its antenna, nor exchange or independently characterise the THz properties of different constituents of the detector, except by performing simulations. Therefore, for comparing different implementations of THz detectors, it is more appropriate to use the power incident on the effective area.

*From the maximal directivity*. Knowing the maximal value of the directivity of the antenna allows for estimating the maximal effective area, which can be associated with the antenna $A_{\mathrm{eff},\mathrm{II}}=D\lambda_{0}^{2}/4\pi$. This does not determine the power delivered into the rectifying device’s circuit but allows for estimation of the power which is incident on the antenna’s effective area. Furthermore, the antenna’s directivity can be both simulated and experimentally assessed, thus providing a reasonable quantity for comparing different devices or implementations.

*Physical area*. The directivity-defined effective areas of individual antennas can sometimes overlap, when the antenna-coupled detectors are arranged in a multipixel imaging array. In this case, the size of the detector pixel can be approximated by the pitch between devices. In Reference [23], arguments have been presented which show that this method, which uses the modeled directivity (as required for method II), predicts the same effective area as the pitch between the pixels. Later in this manuscript, we present experimental evidence that such equality between methods does not apply to devices located at the array’s edges. Furthermore, it must be assured that the devices do not form a dielectric antenna structure. One example of such a case has been reported for a two-dimensional array of detectors that were designed for broadband detection above 600 GHz, which resulted in an unexpectedly enhanced local response at 300 GHz [81]. The physical area is sometimes specified as the lens’s aperture; however, the term "physical" may be confused with the area of the metal [82] or the area of the structure having subwavelength dimensions, as for grating-coupled devices [55,83] or when considering the area of antenna slot [84].

*Area of the diffraction limited spot*. The area of the diffraction-limited spot, $A_{\mathrm{diff}}=\lambda_{0}^{2}/4$, has been often used for devices either without dedicated antennas or by contrast, in order to provide a more conservative estimate, compared to the physical area of elements with the subwavelength dimensions [85]. However, in the referenced paper, the authors defined the diffraction-limited spot area as $A_{\mathrm{diff}}=\lambda_{0}^{2}/\pi$. Moreover, these arguments do not address the involved mechanism of radiation coupling and do not allow for quantitative comparisons, reducing the result to the statement whether the devices are suitable for practical operation.

*Normalised for the omnidirectional antenna case.* There have been reports in the recent literature in which the authors presented the so-called gain or directivity de-embedded device characteristics for the omnidirectional antenna case (i.e., $A_{\mathrm{omni}}=\lambda_{0}^{2}/4\pi$). This area is a factor of π smaller than $A_{\mathrm{diff}}$ and sometimes is interpreted as a circular-shaped diffraction-limited spot; however, the detrimental difference between both methods is that the authors presented either the simulated or experimentally estimated antenna directivity *D* value, thus making the report traceable. However, although this quantity has started to gain popularity in the engineering community, it can hardly improve upon the credibility of later reports as, for now, the main improvement in detector sensitivity comes from the comparison to performance values reported without the de-embedding procedure. This becomes obvious when performing a scrutinous check of the comparison tables reported in [67,69,70].

*Without any normalisation.* There are a wide range of applications, such as raster-scan imaging, spectroscopy, and other systems that exploit point-to-point configuration, which require the detector’s maximal sensitivity, regarding the total power, to be available in the directed (collinear) THz radiation beam. It is worth noting that, although a detector can be adequately characterised using the above-mentioned traceable methods, the optimum THz radiation coupling might require substituting the substrate lens with a different one. In such cases, as well as for direct comparison with the optical performance of calibrated commercially available devices, such as bolometers, Golay cells, pyroelectric sensors, or quasioptically coupled Schottky diode detectors, one can present the performance of optimised devices by referring to the power of the beam (i.e., without imposing any assumptions on the detector area).

Furthermore, there have been reports on different methods which can be used in order to estimate the effective area of the detector; for example, by performing the deconvolution of recorded images (i.e., by numerical analysis of the measured spatial intensity distributions of the THz beam by employing fast Fourier transforms or the Richardson Lucy algorithm) [86,87].

Continuing the discussion about sensitivity estimation methods, it should become evident that all summarised methods that rely on evaluating the effective area can be treated as legitimate, if the authors provide information on the directivity of the detector or an equivalent traceable measurement of the effective area.

In the following, we present a summary of state-of-the-art detector performance values reported for devices implemented using silicon technologies (Table 2), using III–V material systems (Table 3) or involving novel low-dimensional structures (Table 4).

The presented tables allow for the observation that directivity simulations or measurements have been mostly reported for the devices implemented in silicon material systems, which benefit from the most developed fabrication techniques. Reports on detectors which utilise novel low-dimensional materials, in most cases, used the diffraction-spot limited definition to estimate the effective area. The spread of methods used for devices fabricated using III–V materials is also significant. However, there have been reports on the nonscaled optical performance for all systems, clearly demonstrating that the performance of III–V devices is on par with that of silicon devices, while novel materials are also closely approaching state-of-the-art sensitivities.

In the following section, we present a model case for antenna-coupled detector characterisation, which illustrates the challenges arising with regards to choosing the most appropriate method for the performance evaluation. Our experimental characterisation results are compared with the device modelling predictions, allowing the reader to draw independent conclusions.

## 3. Samples and Measurement Setup

### 3.1. Detector Array Design

For the experimental part of this study, we implemented a 2 × 7 array of devices with an identical rectangular microstrip antenna (see Figure 1). Each detector is a patch antenna-coupled N-channel FET fabricated using commercial 65 nm CMOS technology with 9 metal layers (the TSMC foundry, Taiwan). The antenna was implemented in the top metal layer, with dimensions of 110 × 110 µm. The ground plane was formed by combining metallic sheets of the two lowest metal layers. A stack of metals and dielectric layers was formed on top of a slightly conductive silicon substrate (resistivity: 10 Ohm·cm) with a thickness of 279 µm and relative permittivity of 11.9.

Each detector was surrounded by a 6 µm-wide conductive metal wall, formed by all-metal layers and vias between them. The dimensions of the metal cup were 254 × 254 µm. More design details and the principal layout schematic can be found in Reference [19]. In total, we implemented 14 detectors with vertical and horizontal pitch of 279 µm, code-named from $C_{1}$ to $C_{14}$. All devices, located in one row and relevant to this study, employed transistors with the same 450 nm gate width while having individual gate lengths: 60 nm, 80 nm, 100 nm, 120 nm, 150 nm, 200 nm, and 240 nm, with assigned names $C_{8}$ to $C_{14}$. The die, which contained detectors, was packaged using a standard dual-in-line package (DIL) with 40 pins.

### 3.2. THz Characterisation Setup

The experimental schematics employed to characterise the detector frequency and angular response characteristics are presented in Figure 2a. It consisted of a continuous-wave terahertz spectrometer (TeraScan 1550 platform, Toptica [93,94]), which was used as the tunable source for THz radiation. The THz beam was generated by a photomixer, which was excited with two fiber-coupled distributed-feedback diode lasers (*Laser 1* and *Laser 2*), operating around 1.5 µm with a slight difference of individual emission wavelengths. The radiations of both lasers were combined using a 50:50 fiber coupler and converted to the THz frequency range by a self-complementary broadband antenna placed on an InGaAs wafer and employing an electron-hole recombination process [95]. The frequency of the generated THz signal was equal to the laser heterodyne frequency or “beat” [95,96]. The generated signal could be tuned in a wide frequency range, by controlling the temperature and current of both lasers.

We used a set of lasers that allowed for tuning of the photomixer emitted radiation within the frequency band of 100–1200 GHz, with a total output power of 123 µW and 1.25 µW, respectively. The THz radiation was focused on the detectory using two off-axis parabolic mirrors with focal lengths of 2${}^{\prime \prime }$ and 4${}^{\prime \prime }$, as shown in Figure 2b.

The incoming THz radiation had a linear polarisation with Gaussian intensity distribution perpendicular to the detector surface, as shown in Figure 2c. The radiation power level was measured using a calibrated pyroelectric detector [2] with an active area diameter of 20 mm${}^{2}$ placed at the detector plane. The resolution of the beam scan was 0.1 × 0.1 mm${}^{2}$. The intensity $I_{0}$ in the centre of the focal spot can be estimated from the total power, $P_{0}$, as $I_{0}=2P_{0}/\left(\pi w_{0}^{2}\right)$. Here, $w_{0}$ is the radius of the beam taken, concerning the $P_{0}/e^{2}$ level, where $w_{0}=W_{\mathrm{FWHM}}/\sqrt{2\ln2}$, with $W_{\mathrm{FWHM}}$ being the full width at half maximum. According to the measurement, the radiation beam’s diameter at the half-power (–3 dB) point was 1.6 mm (see Figure 2c).

The transistor gate should be biased by a constant potential, in order to set it to the desired work point. An off-chip voltage supply typically provides this, through a bonding wire, pad, and on-chip bus line, which is common for all detectors on one chip. The transistor drain’s output signal was registered using a lock-in amplifier synchronised with the radiation source. The lock-in amplifier (Signal Recovery 7270) had a nominal voltage input noise level of 5 nV/$\sqrt{\mathrm{Hz}}$ at 1 kHz, for an input impedance of 10 MΩ. In our measurement setup, the final noise level did not exceed 360 nV at a modulation of 1 kHz. The XYZ motorised linear stages allowed us to tune the detector’s positioning and analyse the radiation beam distribution.

The left vertical axis of Figure 3 shows the signal-to-noise ratio (SNR) as a function of frequency in power decibels (i.e., $10\cdot \log_{10}\mathrm{SNR}$), while the right vertical axis of the figure represents the integrated power in the radiation beam. The device C${}_{11}$ demonstrated an SNR better than 50 dB with integration time of 100 ms the maximum at 620 GHz for a total power of 7.5 µW in the beam. The “dip” near 557 GHz is the H${}_{2}$O absorption line [97]. The gate was biased to an optimal position of 0.5 V; details of operating point selection are provided in Section 4.1 and Section 5.2.

## 4. Modelling

### 4.1. Circuit Level Modelling of Detector

The modelling of high-frequency signal rectification with FET was performed using the standard transistor model provided by the TSMC foundry and implemented in a circuit simulator, which allowed for harmonic balance analysis. According to the implemented solution, the role of the integrated patch antenna was accounted for by introducing an equivalent power source between the FET source terminal and the ground. This power source had an impedance equal to that of the simulated antenna impedance $Z_{A}$ and the power resulting from the product of half of the impinging power, $P_{0}$, and simulated antenna efficiency $\eta_{a}$ and $\eta_{s}$, as shown in Figure 4a, where $\eta_{s}$ accounts for the absorption efficiency of the receiving antenna. Its value was not simulated but, instead, we used a factor of 0.5, which is typical for half-wavelength dipole antennas [98]; however, depending on the antenna implementation, it may be close to 1 [98,99]. The detector circuit can be operated either in voltage or current readout mode.

As detector modelling requires emulating the antenna as an equivalent power source, in order to account for simulation-relevant antenna characteristics, we employed the principle of reciprocity and performed electromagnetic simulations using the finite-difference time-domain method. The results of these simulations are presented, in detail, in the following section. All selected antennas had a typical radiation efficiency value, $\eta_{a}$, of about 0.5 at 620 GHz. The simulation results for the impedance of antennas and transistors, $Z_{T}$, are shown in Figure 4b,c, respectively. All antennas had very similar impedance characteristics, with variations being lower than 4 Ω. The detector C${}_{14}$ had the best impedance, when matched to the antenna impedance at 620 GHz (close to 1.0); however, the more considerable capacitive contribution of detector C${}_{8}$ also resulted in a good matching factor (close to 0.5; compare the simulated impedance values at 620 GHz in Figure 4b,c).

We should note that, for precise performance evaluation of a terahertz detector, it is necessary to estimate the impinging power, $P_{0}$, which requires the knowledge of the experimentally measurable intensity distribution and the effective area of the receiving antenna. The latter can be experimentally estimated from angular antenna response characteristics or can be deduced from simulations.

### 4.2. Electromagnetic Simulations of Antenna

We decided on microstrip patch antennas for this study. This type of antenna belongs to a well-studied antenna class and has good, predictable radiation properties under front-side illumination. Detectors have been implemented using a similar approach, as detailed in Reference [19]. The patch antenna was implemented in the top metal layer of the CMOS technological process, with dimensions of 110 × 110 µm (see insert in Figure 2a). We applied the finite element method with an adaptive mesh using the CST Studio Suite software, in order to model the investigated antenna structures and evaluate the electromagnetic field’s far-field distributions.

One of the distinctive properties of patch antennas with ground planes is that the lower metal planes allow for efficient suppression of radiation propagation into the thick dielectric substrate. The impact of radiation propagating in the substrate has been addressed in many works which reported about the dependencies of antenna parameters on the substrate thickness [100] and the detector’s position on the substrate [101,102]. Although such a phenomenon can be deliberately used to increase the antenna directivity, it might challenge estimation of the antenna’s effective area. However, the effect may even be used to boost the efficiency of a detector, such as in [46]. To clarify the role of radiation leaking into the substrate in our devices, we investigated two cases numerically. As a metric, we used *D*, the antenna’s directivity, which was used to determine the device sensitivity. The first case was a single device on a finite-size dielectric substrate. The second case corresponded to an exact device model with a double line array having 14 detector elements with identical parameters and a pitch of 279 µm, as shown in Figure 5.

The far-field radiation analysis in the performed simulations demonstrated a difference in the maximum amplitude of the main lobes. In the case of a single device, the main lobe’s amplitude was *D* = 6 dBi, with an angular width of 89, and the radiation direction was perpendicular to the surface. In the array arrangement, we simulated an increase in directivity, to 8.65 dBi for the device on the die edge and 8.45 dBi for devices in the chip’s middle, as shown in Figure 6a. The resulting angular widths were 89 and 95, and the main lobe had a shift of 12 and 9, respectively. Other devices in the middle part of the chip had the same difference as those near the edges.

Figure 6b presents the results of directivity modelling for two detector array elements with substrate thicknesses ranging from 0.1 µm up to 1 mm. Decreasing the substrate thickness demonstrated the relatively stable value of the device’s directivity in the middle part of the slab, as the edge device.

## 5. Experimental Results

### 5.1. Methodology of Measuring Device’s Effective Area and Determining Input Power

The performance of an antenna-coupled THz detector can be presented in several different ways. One of the traceable detector-related metrics is its cross-sectional responsivity, which only considers the amount of power impinging on to the antenna’s effective area, $A_{\mathrm{eff}}$. The effective area, $A_{\mathrm{eff}}$, can be considered in several different ways, which were previously discussed and summarised in Table 1.

We applied a standard methodology for the characterisation of receiving antennas and derived the effective area from the detector’s directivity. The directivity was determined by recording the rectified voltage as a function of tilt angles of the devices in the E- and H-planes. The radiation patterns measured at 620 GHz in these two projections are shown in Figure 7. As can be seen from the figure, for C${}_{9}$–C${}_{11}$ we had an almost 10-degree shift in the angle of maximal gain, which indicates that part of the radiation propagated through the surface or the substrate. The maximal gain angle remained untilted in test simulations for a single antenna with symmetric background and without substrate.

A summary of the experimental results for devices $C_{8}$–$C_{14}$ is presented in Figure 7. For comparison, the modelled antenna’s radiation patterns for the central device C${}_{12}$ in the E- and H-planes are shown as a dashed line in the same figure. As can be seen, the patterns for C${}_{9}$–C${}_{13}$ were similar to each other. Patterns C${}_{8}$ and C${}_{14}$ had a slightly lower shift of maximum direction in the E-plane. The simulation results for C${}_{12}$ were in good agreement with the measured pattern.

Table 5 gives the experimentally determined FWHM angles for C${}_{8}$–C${}_{14}$ at 620 GHz. The maximum directivity can be evaluated after extracting the values of FWHM angles $\Theta_{E}$ and $\Theta_{H}$, employing the expression for the directivity *D* in the approximation for small-angle values: $D\;\approx \;72815/(\Theta_{H}^{2}+\Theta_{E}^{2})$ (for more details, see [48], p. 54). We can use *D* for estimation of A${}_{eff}$, as shown in the remarks for method II in Table 1. We note the excellent quantitative agreement at 620 GHz between the directivity values estimated with this equation and those obtained from the experimental data. The simulations were also performed for a single detector. We obtained a directivity of 6 dBi, giving an effective area of 0.074 mm${}^{2}$, which was quite close to the chosen pitch area (0.076 mm${}^{2}$).

As shown in Table 5, the calculated effective area for detectors placed in an array resulted in slightly larger values than for a single detector. We also measured the dependency of the response for one detector (C${}_{8}$, which was the most sensitive one, due to the shortest gate length) at different frequencies of incoming THz radiation (590–660 GHz) in both the E- and H-planes. The results are shown in Figure 8. As can be seen from the figure, the angle of maximum gain for C${}_{8}$ at different frequencies varied, implying the existence of propagating substrate waves.

With the detector effective area values estimated through the simulated and presented experiments, we then obtained estimates for the impinging power within the effective antenna cross-section, using $P_{in}=I_{0}\cdot A_{\mathrm{eff}}$.

### 5.2. Terahertz Responsivity

Figure 9 shows the measured cross-sectional voltage ${ℜ}_{V,c}$ (left) and current responsivity ${ℜ}_{I,c}$ (right), together with simulation results (dash line) obtained with the TSMC 65 nm model from the process technology provider. Using an experimentally defined effective area, we obtained a maximum cross-section voltage responsivity of 1414 V/W for C${}_{8}$ at a gate bias voltage of $V_{G}$ 0.35 V, which was below the device’s threshold voltage. The peak of current responsivity was 66 mA/W, corresponding to a $V_{G}$ of 0.65 V. As can be seen, a good correlation between the simulation and measurement results was obtained. The slight difference between the foundry model-predicted responsivity and experimental data could be attributed to the slight scattering of individual device parameters, whereas the simulation was performed using an average representative. For comparison, we also present the estimation of cross-section responsivity expected for a single detector with D = 6 dBi, under substrate wave suppression. As can be seen in Figure 9, the value may become about 2 times larger than the measured responsivity; however, this would exceed the modelled value. The comparison, thus, underlines the necessity of thorough analysis of array performance, instead of using simplified assumptions such as assuming the area to be defined by the pitch or the effective area of the idealised patch without the manifestation of substrate waves.

The used figure-of-merit, which defines the detector’s sensitivity, was the NEP. This is defined as the input power that results in a signal-to-noise ratio of one for the equivalent noise bandwidth of 1 Hz. One can calculate the cross-sectional NEP of the transistor, according to $\mathrm{NEP}=N_{V}/{ℜ}_{V,c}$, where $N_{V}$ is the noise voltage spectral density of the detector (or, $\mathrm{NEP}=N_{I}/{ℜ}_{I,c}$, where $N_{I}$ is the noise current spectral density of the detector). As the detection was studied under zero drain bias, thermal noise was the only source of noise for the transistor, which can be calculated by $N_{V}=\sqrt{4k_{B}TR_{\mathrm{DC}}}$, where $k_{B}$ is the Boltzmann constant, *T* is the temperature, and $R_{\mathrm{DC}}$ is the DC drain-source resistance of the transistor. This statement has been tested on many different devices of the same kind and has been addressed in a series of reports [24,25,103].

The measured and simulated NEP values of C${}_{8}$ under different gate bias conditions at a radiation frequency of 620 GHz are plotted in Figure 10. The minimum measured cross-sectional NEP was 19.2 pW/$\sqrt{\mathrm{Hz}}$, corresponding to a V${}_{G}$ of 0.59 V. However, this value can get as low as 10 pW/$\sqrt{\mathrm{Hz}}$ assuming D = 6 dBi, which was even slightly below the predictions by simulations, which are presented with a red line. A minimum simulated value of 13.4 pW/$\sqrt{\mathrm{Hz}}$ at a $V_{G}$ of 0.55 V was obtained. Therefore, we can conclude that our estimated performance values well-matched those predicted by the simulations, supporting the directivity-related estimation of the effective area. It is important to note that the only assumptions we used in model simulations were related to the antenna efficiency factors, and only these were taken from the results of the antenna’s electrodynamic modelling. The slight differences in the shapes of the curves might have originated from the fact that the simulation model described statistically averaged device characteristics, whereas the experiments were performed using individual devices.

Table 6 compares the performance of 65 nm CMOS detectors, considering the different methods used to calculate the effective area. As one can see from the table, considering the effective area to be equal to the physical area (pitch between devices) or as the area of the diffraction-limited spot resulted in a smaller estimated effective area, compared to the experimental estimate and led to about two-fold overestimation of the performance of the sensor (i.e., increasing in the responsivity or decreasing in the value of NEP). Due to the selection of focusing mirrors, the optical performance (presented in the last line) was about 20-fold lower than the cross-sectional performance. The main reason for this lies in the used optics, with much lower numerical aperture, which did not match the antenna’s angular radiation characteristics. Nevertheless, another conclusion can be drawn, regarding the meaning of 2.5 pW/$\sqrt{\mathrm{Hz}}$, which would result as the NEP for the omnidirectional antenna case. As it was not efficiently de-embedded and was much lower than the expected value from our simulations (of 13.4 pW/$\sqrt{\mathrm{Hz}}$), as expected without any assumption of area, it has no direct relevance to either the electrical or optical performance of the devices, besides being embedded again with the directivity value.

## 6. Conclusions

In this paper, we presented a review of the methods currently in use for the characterisation of antenna-coupled field-effect transistor-based THz detectors. We showed that at least six primary characterisation methods are commonly applied. While all of these methods are valid in the range of their applicability, in most cases, it can become challenging to select a correct method without performing thorough experimental characterisation, which is required to assess the effective area of the detector. We present a comprehensive study of field-effect transistor-based antenna-coupled detectors at a frequency of 620 GHz, in order to illustrate the ambiguities arising in practical cases. Based on numerical simulations, supported by experimental characterisations of the angular response dependence, we showed that the detector’s effective area in a small array can become considerably larger than that defined by the pitch between devices or when considering a single device. For the resonant frequency of 620 GHz and by considering the maximum directivity-defined effective area, we arrived at a room-temperature cross-sectional noise-equivalent power of 19.2 pW/$\sqrt{\mathrm{Hz}}$. In contrast, for the omnidirectional antenna, this value was reported as low as 2.5 pW/$\sqrt{\mathrm{Hz}}$; however, the latter value was not directly linked with either the optimal or achievable performance and, for the concomitant system, the analysis should be re-embedded with the measured directivity value of 8.7 dBi. This example shows how, by mixing between methodologies, one can manipulate the NEP to one order of magnitude in overestimating the detector’s performance and seemingly breaking the 10 pW/$\sqrt{\mathrm{Hz}}$ barrier.

Finally, we wish to summarise that, when comparing the performance of different devices or novel detection methods, the evaluation method used should be based on the directivity (or by any other technique with a traceable assessment of the effective sensor area). The presented results detailed sensitivity classification methods paving the way towards the unified scientific metrology of FET-based THz sensors, which is important for both researchers competing for records as well as potential users and engineers who intend to utilise the sensors to construct future THz systems.

## Figures and Tables

**Figure 1 sensors-21-02909-f001:**
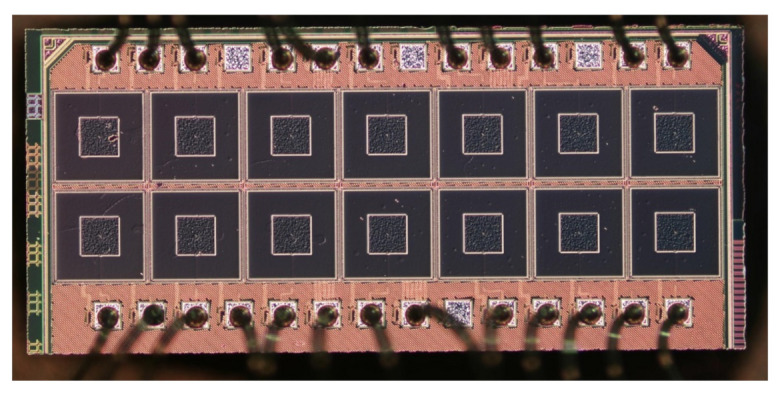
Micrograph photo of the die with detector array (the pitch in both directions is 279 µm).

**Figure 2 sensors-21-02909-f002:**
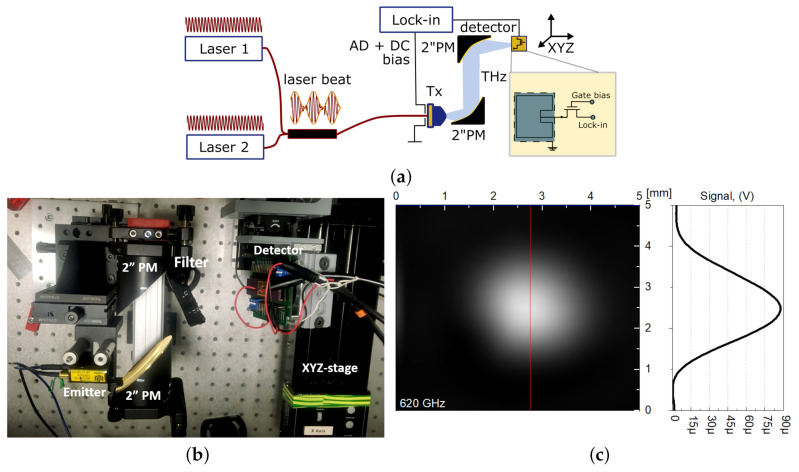
(**a**) Schematic building blocks and (**b**) photo of a CW THz characterisation system; (**c**) 2D scan of beam intensity at a detector location for 620 GHz.

**Figure 3 sensors-21-02909-f003:**
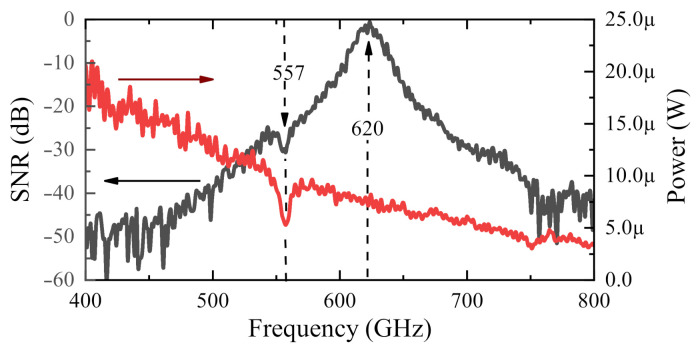
The black line shows the signal-to-noise ratio (**left** axis) in decibels of power (i.e., $10\cdot \log_{10}\mathrm{SNR}$) for the detector C${}_{11}$ as a function of frequency. The red line shows the integrated power of the THz radiation source (**right** axis).

**Figure 4 sensors-21-02909-f004:**
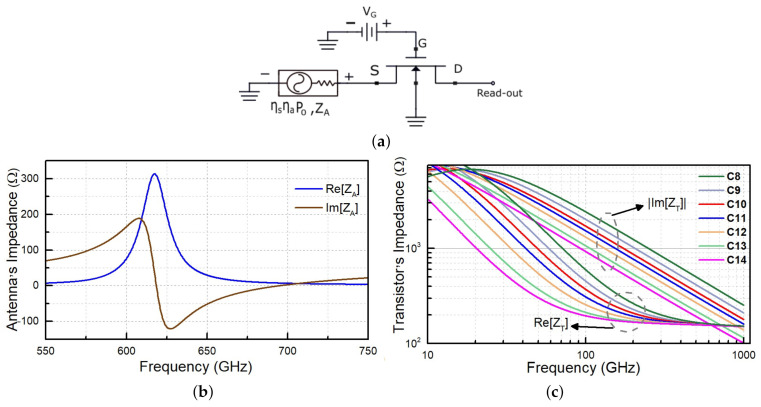
(**a**) Equivalent circuit of field-effect-transistor based detector; (**b**) Simulated patch antenna’s impedance, $Z_{A}$, versus frequency for C${}_{11}$; and (**c**) Real and imaginary parts of simulated transistor’s impedance, $Z_{T}$, versus frequency for the gate bias corresponding to the optimum operation point for C${}_{8}$–C${}_{14}$.

**Figure 5 sensors-21-02909-f005:**
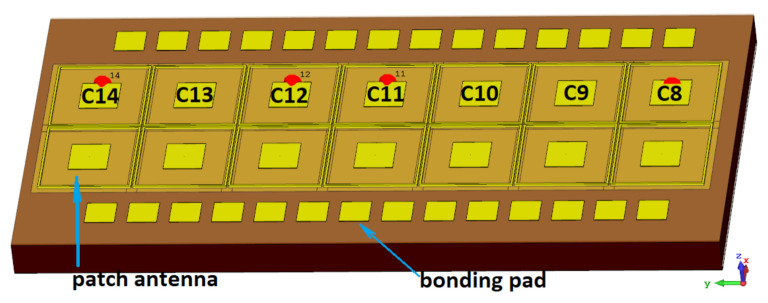
A 3D model of a die with a 2 × 7 array of identical antennas and contact pads on a Si substrate.

**Figure 6 sensors-21-02909-f006:**
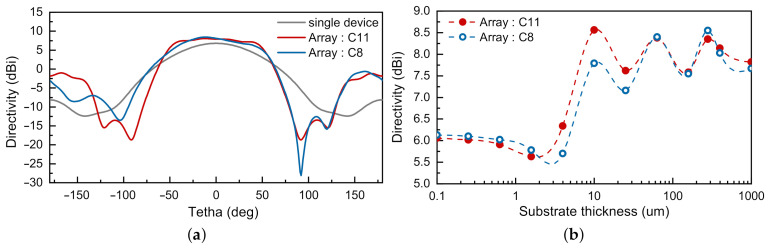
(**a**) Comparison of radiation patterns in the E-plane for a single-chip device (grey line) and a device integrated into the array. The blue line corresponds to a detector positioned in the middle of the array line (C${}_{11}$), while the red line is for the device placed at the side (C${}_{8}$). The result is presented for 620 GHz; (**b**) Directivity as a function of substrate thickness for the device in the middle of the array (red line) and at the side position (blue line).

**Figure 7 sensors-21-02909-f007:**
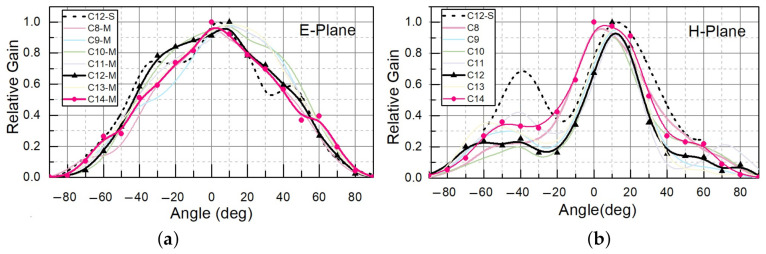
Dependency of the THz response of C${}_{8}$–C${}_{14}$ for various (**a**) E-plane and (**b**) H-plane tilt angles at 620 GHz. The simulated response of C${}_{12}$ is displayed with a dotted line.

**Figure 8 sensors-21-02909-f008:**
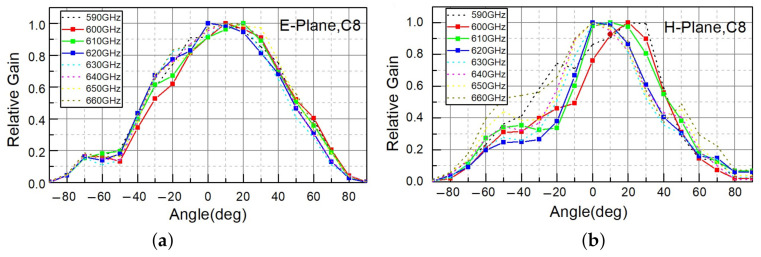
Dependency of the THz response of C${}_{8}$ on (**a**) E-plane and (**b**) H-plane tilt angles for different incoming THz radiation frequencies.

**Figure 9 sensors-21-02909-f009:**
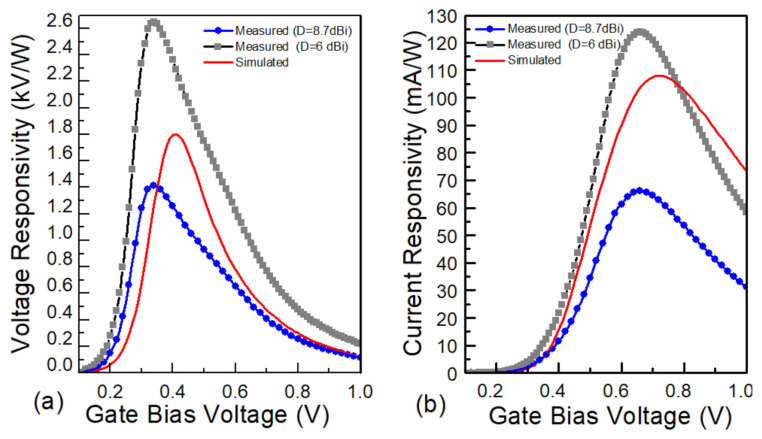
Voltage (**a**) and current (**b**) responsivity of C${}_{8}$ vs. gate voltage bias at 620 GHz. Two different directivity values were used for the responsivity estimation: D = 6 dBi (the detector treated as a single one, the gray line) and D = 8.7 dBi (treated as a part of an array, the blue line). The red line shows the simulation results obtained with the TSMC 65 nm model.

**Figure 10 sensors-21-02909-f010:**
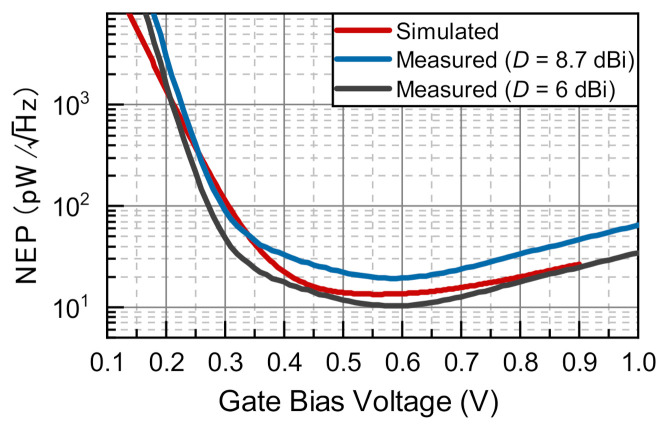
Noise equivalent power of C${}_{8}$ as a function of the gate voltage bias.

**Table 1 sensors-21-02909-t001:** Summary of methods used for estimation of the effective area of THz detectors.

Method for calculation of $A_{\mathrm{eff}}$	Comment	Ref.
I. From the antenna gain	$A_{\mathrm{eff}}=\frac{G\lambda_{0}^{2}}{4\pi }$ This is a widely accepted method in the field of microwave antennas, which has been described in textbooks and simulation tools. It accounts for the power loss due to antenna efficiency and allows for estimating the power which is applied to the detector circuit.	[34,35,36,37], [38] (pp. 1–10)
II. From the maximal directivity	$A_{\mathrm{eff}}=\frac{D\lambda_{0}^{2}}{4\pi }$ This definition describes the maximum antenna effective aperture. Therefore, it is best suited for the determination and comparison of optical characteristics of detectors.	[17,19,24,39,40,41,42,43,44,45,46,47], [48] (p. 92)
III. Physical area	In multipixel imaging arrays with overlapping effective areas, the detector pixel size can be approximated by the pitch between devices. However, it must be assured that the devices do not form a dielectric antenna structure. In some works, the physical area has been specified as the aperture of the substrate lens.	[9,10,11,23,39,49,50,51,52,53,54,55,56,57]
IV. Area of the diffraction-limited spot	$A_{\mathrm{diff}}=\frac{\lambda_{0}^{2}}{4}$ This area is often used for devices without dedicated antennas. It has been claimed to be a conservative estimate; however, in most cases, it leads to the substantial overestimation of detector performance.	[31,58,59,60,61,62,63,64,65]
V. Normalised for the omnidirectional antenna case	$A_{\mathrm{omni}}=\frac{\lambda_{0}^{2}}{4\pi }$ This area is used for devices with known (simulated or measured) directivity, by normalising it to unity for the omnidirectional antenna case (in other words, the antenna gain is de-embedded) and, for devices without dedicated antennas, it is interpreted as a circular-shaped diffraction-limited spot.	[28,66,67,68,69,70]
VI. Without any normalisation	The optical performance without normalisation of the incident power is relevant for a wide range of applications, such as raster-scan imaging, spectroscopy, and other systems exploiting point-to-point configurations. The performance values obtained in this way can be compared with that of commercially available devices, such as bolometers, Golay cells, pyroelectric sensors, or quasioptically coupled Schottky diode detectors.	[26,29,71,72,73,74,75,76,77,78,79,80]

**Table 2 sensors-21-02909-t002:** Summary of reports on silicon-based THz detectors.

Technology	Freq.	Antenna	NEP	Responsivity	Single or Array	Method-Ology *	Ref.
	**GHz**		**pW/** $\sqrt{\mathrm{Hz}}$	**V/W**			
MOSFET, 90 nm CMOS	250	Slot + Si lens	21	408	Single	VI	[80]
MOSFET, 90 nm CMOS	250–750	Various	40	185 k **	Single	II	[45]
HBT, 130 nm SiGe	292	Wire ring + Si lens	1.9	9 k	Single	V	[67]
MOSFET, 90 nm CMOS	300	Slot + Si lens	20.8	55 k **	Single	VI	[88]
MOSFET, 90 nm CMOS	300–1500	Bow-tie + Si lens	48–70	45	Single	VI	[26]
MOSFET, 65 nm CMOS	315	CSDRA	3.5	2 k	Single	II	[46]
HBT, 130 nm SiGe	430	Wire-ring +Si lens	2.7	5 k	Single	V	[68]
MOSFET, 90 nm CMOS	590	Patch	20	-	Single	II	[24]
MOSFET, 150 nm CMOS	595	Patch	42	350	Single	II	[17]
MOSFET, 130 nm CMOS	600	Bow-tie	25.9	216 k **	Array 31 × 31	III	[9]
MOSFET, 150 nm CMOS	600	Patch	43	300	Array 24 × 24	III	[10]
MOSFET, 22 nm FD-SOI CMOS	605	Double-folded dipole + Si lens	2.3	32 k	Single	V	[69]
MOSFET, 65 nm SOI CMOS	650	Folded dipole + Si lens	17	1930	Array 3 × 5	III	[51]
MOSFET, 130 nm SiGe BiCMOS	650	Ring + Si lens	80	450	Single	III	[50]
MOSFET, 250 nm CMOS	650	Patch	300	80 k **	Array 3 × 5	III	[53]
HBT, 250 nm SiGe	700	Ring + Si lens	50	1 A/W	Array 3 × 5	II	[39]
MOSFET, 65 nm Si CMOS	724	Ring + Si lens	14	2200	Single	III	[49]
P-N diode, 45 nm CMOS	781	Patch	56	558	Single	II	[43]
Diode-connected MOSFET, 130 nm CMOS	823	Patch	36.2	2560	Array 8 × 8	III	[52]
MOSFET, 22 nm FD-SOI CMOS	855	Ring + Si lens	12	0.180 [A/W]	Single	I	[37]
MOSFET, 180 nm CMOS	860	Patch	106	3300	Array 3 × 5	II	[40]
SBD, 130 nm CMOS	860	Patch	42	273	Single	II	[41]
MOSFET, 65 nm CMOS	1000	Bi-quad + Si lens	25	765	Single	VI	[25]
MOSFET, 90 nm CMOS	2520 3110 4250	Patch	63 85 110	336 308 230	Single	II	[8]
MOSFET, 65 nm CMOS	3000	Patch	73	526	Array 12 × 9	II	[47]
**MOSFET, 65 nm CMOS**	**620**	**Patch**	**19.2**	**1400**	**Array 2 × 7**	**II**	**This work**

* See Table 1. ** Amplified read-out was used for calculation of responsivity.

**Table 3 sensors-21-02909-t003:** Summary of reports on III–V THz detectors.

Technology	Freq.	Antenna	Min. NEP	Responsivity	Single or Array	Method-Ology	Ref.
	**GHz**		**pW/** $\sqrt{\mathrm{Hz}}$	**V/W**			
GaN HEMT	140	Nano-antenna	0.58	15.5 k	Single	III **	[89]
DGG-HEMT, InAlAs/ InGaAs/InP	200	Grating coupling	0.48	22.7	Single	III ${}^{*}$	[83]
GaAs HEMT	271, 632	-	135, 1250	42, 1.6	Single	II	[42]
AlGaN/GaN SSD	300	-	280	100	Array	VI	[75]
GaAs HEMT	300	Dipole	9.1	8.5 k	Single	III ${}^{‡}$	[82]
GaAs/AlGaAs FET	305	-	1330	11	Single	VI	[76]
AlGaN/GaN HEMT	490–645	Bow-tie + Si lens	25–31	104 [mA/W]	Single	VI	[72]
InGaAs/AlGaAs HFET	592	Bow-tie + Si lens	500	20	Single	III	[56]
AlGaAs/GaAs HEMT	600	Log-spiral + Si lens	250	20–40 [mA/W]	Single	VI	[90]
AlGaN/GaN HEMT	700–925	Assym. dipole + Si lens	30	-	Single	III ${}^{†}$	[91]
AlGaN/GaN HEMT	897	Assym. dipole	40	3.6 k	Single	III ${}^{†}$	[57]
AlGaN/GaN HEMT	900	Bow-tie + Si lens	57	48 [mA/W]	Single	VI	[74]
DGG-HEMT, InAlAs/InGaAs/InP	1000	Grating coupling	15	2.2 k	Single	III ${}^{*}$	[55]
InGaAs/GaAs	1630	-	$1\times 10^{5}$	170	Single	VI	[79]

* Normalised to subwavelength dimensions of 20 × 20 µm${}^{2}$. ${}^{†}$ The area of 200 × 200 µm${}^{2}$ is stated only in Ref. [57]. ${}^{‡}$ Normalised to the physical area of metal in the antenna. ** Normalised to subwavelength dimensions of 15 × 35 µm${}^{2}$.

**Table 4 sensors-21-02909-t004:** Summary of reports on THz detectors employing novel low-dimensional materials.

Technology	Freq.	Antenna	Min. NEP	Responsivity	Single or Array	Method-Ology *	Ref.
	**GHz**		**pW/** $\sqrt{\mathrm{Hz}}$	**V/W**			
GFET	130–450	logarithmic spiral	600	20	Single	VI	[77]
BL-GFET	290–380	–	2000	1.2	Single	IV	[59]
BP-based FET	300	Bow-tie	$1\times 10^{4}$	-	Single	IV	[63]
BP-based FET	300	Bow-tie	$4\times 10^{4}$	0.15	Single	IV	[65]
DGG-GFET	300	-	$9\times 10^{5}$	-	Single	IV	[60]
Graphene Ballistic Rectifier	300	Bow-tie	34	764	Single	V	[28]
Nanowire-FET	300	Bow-tie	$2.5\times 10^{3}$	1.5	Single	IV	[62]
Nanowire-FET	300	Bow-tie	1000	100	Single	IV	[64]
SL-GFET, BL-GFET	300	a log-periodic circular-toothed	$2\times 10^{5}$ (SLG) $3\times 10^{4}$ (BLG)	-	Single	IV	[61]
GFET	400	Bow-tie	130	74	Single	VI	[71]
GFET	487	Bow-tie	3000	2	Single	VI	[92]
GFET	600	Bow-tie	515	14	Single	VI	[29]
Graphene	2000	Bow-tie	$15\times 10^{4}$	-	Array	VI	[78]
GFET	2800	Bow-tie	160	-	Single	IV	[31]

* See definitions given at the Table 1.

**Table 5 sensors-21-02909-t005:** Summary of the directivity measurements and simulation.

	$\Theta_{E}$	$\Theta_{H}$	*D*	$A_{\mathrm{eff}}$	*D*	$A_{\mathrm{eff}}$
Detector			Meas.	Meas.	Sim.	Sim.
	(deg)	(deg)	(dBi)	(mm${}^{2}$)	(dBi)	(mm${}^{2}$)
C${}_{8}$	87	47	8.70	0.138	8.65	0.136
C${}_{9}$	93	38	8.58	0.134	8.43	0.130
C${}_{10}$	94	35	8.59	0.134	8.65	0.136
C${}_{11}$	93	32	8.76	0.140	8.45	0.130
C${}_{12}$	93	36	8.60	0.136	8.73	0.139
C${}_{13}$	90	38	8.82	0.142	8.59	0.134
C${}_{14}$	85	46	8.90	0.145	7.83	0.113

**Table 6 sensors-21-02909-t006:** Comparison of responsivity and NEP of the detector, considering different methods calculating the effective area.

Method	$A_{\mathrm{eff}}$	Max. Responsivity	Min.NEP
	(mm${}^{2}$)	(V/W)	(pW/$\sqrt{\mathrm{Hz}}$)
Measured directivity of the detector in array *D* = 8.7 dBi)	0.138	1414	19.2
Simulated directivity of single detector *D* = 6 dBi	0.074	2651	10.28
Physical area defined by the pitch between devices	0.077	2545	10.7
Area of diffraction limited spot	0.058	3368	8.09
From the antenna gain ($G=\eta_{a}\cdot D$)	0.069	2828	9.6
Normalised for the omnidirectional antenna case	0.018	10.8 K	2.5
Without any normalisation (Optical performance)		63.6	428

## Data Availability

The data that support the findings of this study are available from the corresponding author (E.J.) upon reasonable request.

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
