# Peer review of "Sensitivity of Field-Effect Transistor-Based Terahertz Detectors"

_sensors, 2021, doi:10.3390/s21092909_

Round 1

Reviewer 1 Report

 In this review article, the authors discuss the sensitivity of Terahertz (THz) detectors, particularly based on field-effect-transistors (FET). While other technologies used for THz detection are briefly mentioned the introduction, various methods for evaluating the detection sensitivity with special attention to the estimation of the sensor area are extensively reviewed. Different definitions used for the area in the context of FET based detectors for THz detection are described and the associated drawback leading ot he over estimation of the sensitivity are highlighted. Literature surveys are tabulated for easy access and clarity. Further, the experiments conducted by the authors are presented together with simulations. The sensitivity performance characteristics of some of the detectors fabricated by the authors are discussed. This review manuscript can be accepted for publications after the following issues are addressed:

  1. In the introduction, better perspective and context could be provided by discussing about the sub-millimetre size of the antenna required in the THz wavelength range, which is an advantage.
  2. How do the values of the sensitivity, specifically NEP of FET based detectors compare with other sensor technologies such as Goley cells, pyroelectric detectors and bolometers?
  3. On a general pictorial representation of a FET based sensor/sensor array such as the one presented in Fig. 5, indicating antenna/patch will be helpful for the reader.
  4. How is the difference between the two lasers used for the characterization of the sensors controlled? Using the temperature of the diodes? How stable in the THz frequency? Does it have any effect on the sensitivity?
  5. Details of the THz transmitter used for generating the THz radiation should be included.
  6. What is the directivity (D) of the simulations presented in Fig. 9.
  7. Any strategies to minimize propagation through the substrate?
  8. Can the authors comment on the NEP dependence on the frequency in comparison with other detectors?
  9. Inspite of the artificially boosted estimates of the NEP in the literature, how does the results presented in this manuscript compare the state of the art values?
  10. Details about how the simulations were performed should be included.

Minor:

  1. Thorough reading of the manuscript for clarity and typos will be useful.
  2. Panel labels for Fig. 9 are missing.

Reviewer 2 Report

This manuscript reports a study of fifield-effect-transistor-based antenna-coupled detectors for 620 GHz frequency. This work gives a detailed classification method of the field-effect transistor terahertz detector sensitivity. Overall,the work is practical and can be accepted after minor revision. Regarding this, there are some concerns, questions and suggestions as follow:

  1. The English language should be carefully revised throughout the manuscript. Some parts of the paper cannot be understood easily. For example, “This paper presents an overview of different methods used for sensitivity, i.e., responsivityand noise equivalent power determination of state-of-the-art fifield-effect-transistor-based THz detectors/sensors – showing that the presented result may depend very much on the sensor effective area determination method often leading up to orders of magnitude discrepancy.” , “Which challenges might arise when selecting a proper method for characterization is demonstrated with the example of a 2 × 7 detector array utilizing fifield-effect-transistors and monolithically integrated patch antennas for 620 GHz.”
  2. The diagrams in the text should be placed between the text paragraphs and appear in order, for example, Figure 4c appeared before Figure 4a.
  3. The labels of the figures can be changed to the upper left corner for easy viewing.
  4. The conclusion should emphasize the key content and results of the article research, and does not need to enumerate all aspects involved in the article.
